# Understanding the social drivers of antibiotic use during COVID-19 in Bangladesh: Implications for reduction of antimicrobial resistance

Abul Kalam[1]◉*, Shahanaj Shano[2,3]◉, Mohammad Asif Khan[4], Ariful Islam[3], Narelle Warren[5], Mohammad Mahmudul Hassan[6], Mark Davis[5,7]

1 Bangladesh Country Office, Helen Keller International, Dhaka, Bangladesh, 2 Institute of Epidemiology, Disease Control and Research (IEDCR), Dhaka, Bangladesh, 3 EcoHealth Alliance, New York, New York, United States of America, 4 Chattogram Civil Surgeon Office, Chattogram, Bangladesh, 5 School of Social and Political Sciences, Monash University, Melbourne, Victoria, Australia, 6 Chittagong Veterinary and Animal Science University, Chattogram, Bangladesh, 7 Centre to Impact Antimicrobial Resistance, Monash University, Melbourne, Victoria, Australia

◉ These authors contributed equally to this work.
* a.kalam724@gmail.com

**Data Availability Statement:** All relevant data are within the paper and its Supporting information files.

## Abstract

Antimicrobial resistance (AMR) is a global public health crisis that is now impacted by the COVID-19 pandemic. Little is known how COVID-19 risks influence people to consume antibiotics, particularly in contexts like Bangladesh where these pharmaceuticals can be purchased without a prescription. This paper identifies the social drivers of antibiotics use among home-based patients who have tested positive with SARS-CoV-2 or have COVID-19-like symptoms. Using qualitative telephone interviews, the research was conducted in two Bangladesh cities with 40 participants who reported that they had tested positive for coronavirus (n = 20) or had COVID-19-like symptoms (n = 20). Our analysis identified five themes in antibiotic use narratives: antibiotics as 'big' medicine; managing anxiety; dealing with social repercussions of COVID-19 infection; lack of access to COVID-19 testing and healthcare services; and informal sources of treatment advice. Antibiotics were seen to solve physical and social aspects of COVID-19 infection, with urgent ramifications for AMR in Bangladesh and more general implications for global efforts to mitigate AMR.

## Introduction

Along with its direct impacts on the health of individuals, the COVID-19 pandemic is implicated in the increased, off-prescription use of antimicrobials, practices that are thought to contribute to antimicrobial resistance (AMR) [1, 2]. Although clinical services in affluent countries have reported decrease in antibiotic prescription since the pandemic [3], Lower-and-Middle Income Countries (LMICs) are reporting antibiotic use coupled with a higher burden of COVID-19 infection [4]. AMR will further burden healthcare systems and worsen

**Funding:** This work is partially funded by Bangladesh Bureau of Educational Information and Statistics (BANBEIS), ID number SD2019967. Mohammad Mahmudul Hassan was supported by BANBEIS. The funders had no role in study design, data collection and analysis, decision to publish, or preparation of the manuscript.

**Competing interests:** The authors have declared that no competing interests exist.

health outcomes [1, 5]. According to World Health Organization (WHO) interim guidelines, antibiotics can be used when there is secondary co-infection seen among patients with COVID-19 [6], but are discouraged amongst those with mild symptoms. Despite these recommendations, the WHO has reported that azithromycin is widely used with other medicines, and oseltamivir and lopinavir/ritonavir are being combined with antibiotics. Antibiotic use appears to be high among COVID-19 patients [7], with one report indicating up to 45% of coronavirus patients were receiving antibiotic treatment [8]. Clinically-unjustified use of antibiotics is thought to be linked with AMR, which is estimated to claim death of 700,000 people per annum [9]. It is projected that AMR-related mortality will increase during the pandemic and some have reported more than 130,000 deaths in 2020 alone [10]. The factors that drive antibiotic use during COVID-19 need to be documented and understood to strengthen public health efforts to simultaneously respond to the pandemic and reduce AMR [5, 10–14].

LMICs face the twin burdens of increased risk from COVID-19 and AMR. Salient factors include poorly resourced health and hygiene programs, health care services, health governance, and ineffective regulatory and legislative mechanisms controlling antibiotic use [15–17]. Inadequate testing capacity, limitations in health care service provision, and poor health infrastructure are considered major factors for the increased burden of COVID-19 in LMICs [4, 18, 19]. Like many other LMICs, COVID-19 and AMR pose immense threats in Bangladesh. Over the counter (OTC) sales promote easy access to antibiotics [17, 20] through drug shops, private and public health centers, and specialized hospitals [21]. Along with professional doctors, unqualified providers in the informal health care sector also prescribe antibiotics [22, 23]. The aggressive and sometimes unethical marketing practices of pharmaceutical companies are also linked with the over-prescription of antibiotics [24]. Moreover, most antimicrobials are prescribed based on best-guess and without microbial aetiology [23, 25]. However, the Government of Bangladesh (GoB) adopted a National Action Plan (NAP) in 2017 [26] to reduce the impact of AMR. Understanding the socio-cultural dimensions of antibiotic prescription and use has been identified as crucial for the successful implementation of the National Action Plan in Bangladesh [20].

Antibiotic use is deeply embedded in the socio-economic fabric of everyday life in Bangladesh and, in particular, is woven into the significance of community markets for everyday life [27]. Hence, the buying and selling of goods and services shapes the social life of antibiotics [28]. Moreover, taking and giving of medicines contributes to the meaning of healing, over and above chemical action of pharmaceuticals on disease states [29]. Among general populations, the biomedical properties of antimicrobials are not well understood but they are perceived as highly effective and powerful treatments. Research among Australian ethnic communities and the general public showed that people have diverse understandings of antibiotics and AMR, not all of which were aligned with scientific knowledge and authorized recommendations [30–32]. Qualitative research in Malaysian lower income communities noted that antibiotics were understood to hasten recovery from illness, reduce fever and relieve pain [33]. Several New Zealand studies have reported limited knowledge and understanding of antibiotic use [34–37]. A mixed methods study with community members in Pakistan found that antibiotics were perceived as able to cure all types of infections [38]. Several Bangladesh studies reported that antibiotics were perceived as powerful medicines which lead to quick results, work against almost all diseases, including viruses, cold cough, diarrhea, food poisoning, infection, dental carries and toothache, irritable bowel syndrome, acne, ear and throat pain and work as preventative medicines [20, 39–41].

It is important not to frame these research findings as general public's knowledge deficits that can be addressed with more accurate and copious information. Medical anthropologists and sociologists urge for deeper understanding of the beliefs and meanings associated with

antimicrobials, rather than a focus on measuring and correcting knowledge [42, 43]. Explanations of antibiotics and antibiotic use may diverge from biomedical knowledge due to socio-economic status and cultural context [31, 43]. This approach is all the more important in the context of COVID-19, when health and socio-cultures are imbued with new and urgent risks to life.

Our paper, therefore, seeks to provide deeper insight into the social drivers of antibiotic use during the COVID-19 pandemic, in order to inform public health policy and communications on the pandemic and its intersections with AMR. In what follows, we assume that taking and giving medicines are social acts that are deeply imbued with ramified psychological and socio-cultural meanings, perhaps most keenly in time of pandemic. We seek to draw out the explanations and assumptions respondents made about their COVID-19 diagnosis or suspected infection and in that context how they made use of, and justified, antibiotics. Prior to explaining our methods, we provide below some background information about the pandemic in Bangladesh to help contextualize the experiential narratives we analyze in what follows.

## COVID-19 in Bangladesh

Bangladesh reported its first laboratory-confirmed SARS-CoV-2 case on 8[th] March of 2020. As part of the response mechanism, GoB called a nationwide 'lockdown' from 26 March to 30 May. Despite these actions, at the time of the current study (by 29 June 2020), Bangladesh reported 141,801 diagnoses of COVID-19 and 1782 deaths [44]. Misunderstandings of the concepts of general leave and lockdown [45] combined with structural barriers to physical distancing and quarantining [46] compromised the efforts to contain the spread of infection. The extreme insufficiency of testing capacity due to inadequate quantities of testing kits and suitably equipped laboratory facilities suggest that many cases remained undetected. At 29 June 2020, Bangladesh had a test rate of less than 8 in every one hundred thousand [4].

Bangladesh also faced severe challenges in terms of managing confirmed cases in hospital settings due to infrastructural and staffing issues. There are only 399 Intensive Care Units (ICUs) in government hospitals in Bangladesh, with 218 in Dhaka city alone [45]. The size of the health work force is a major problem in Bangladesh, with only 4.90 registered physicians and 2.90 registered nurses per 10,000 population [47]. Moreover, medical workers were provided with low quality personal protective equipment [45], placing them, their patients and attendants at risk of infection. For these reasons, many patients preferred to remain at home fearing lack of access to comprehensive treatment in hospitals [48]. Although the government provided advice for home-based patients, specialists and researchers raised concerns about improper coordination based on the interim guidelines on home care for patients with COVID-19 [49].

## Materials and methods

### Study design

This study employed an exploratory qualitative approach to identify the social drivers of antibiotic use among at home COVID-19 patients and those who were undiagnosed and experiencing symptoms. We explored individual's explanatory models of antibiotics in order to deepen knowledge and understanding of antibiotics use in the age of COVID-19. Where applicable, the study followed the Consolidated Criteria for Reporting Qualitative Research (COREQ) checklist for qualitative research [50].

## Study setting and inclusion criteria

The study was conducted from May to June 2020 in two Bangladeshi cities—Dhaka, the capital city with a population over 21 million, and Chattogram, the largest commercial city with a population of more than 3.9 million. These cities were selected as they were considered epicenters for the spread of COVID-19 [51] and had the highest number of positive cases (Dhaka-71% and Chattogram-14% out of 141,801 total positive cases) at the time of data collection [44]. Potential participants were invited to take part in the interviews if they met the following inclusion criteria: i) they reported that they had been tested positive with COVID-19 or had COVID-19 like symptoms (like fever, dry cough, sore throat and loss of taste or smelling sensation) ii) received home care; and iii) had taken antibiotics during illness.

## Recruitment process and sample size

A purposive sampling was adopted to ensure an even distribution of gender and age. Participants were recruited through social media advertisements, and this was supplemented by a network sampling strategy. People interested in the study contacted the lead researchers (MAK, SS, MMH, AI) who provided a verbal description of the study. During this initial discussion, the potential participants were asked if they had been clinically diagnosed with COVID-19, had symptoms and if they recalled the name of antibiotics they had consumed. All of the participants mentioned the brand names of the antibiotics and these brand names were then linked back to the antibiotic type (group). If a volunteer met the inclusion criteria, they were invited to participate in the interview. Due to COVID-19 safe requirements, those who wished to participate in the study, provided verbal consent before being interviewed by mobile phone by two research assistants, one male and one female. Five individuals refused to take part in the interviews due to their busy schedule.

Interviews continued until thematic saturation was observed. During data collection, geographical location and participants' characteristics (i.e., gender, age, educational attainment) were considered in relation to the inclusion criteria. Forty interviews were conducted in this study. Twenty respondents who reported that they returned a positive COVID-19 test and 20 who had COVID-19 like symptoms.

## Conduct of the interviews

A semi-structured interview guide was used to explore people's explanations for using antibiotics. The guide was developed based on an extensive review of research and review articles, viewpoints, opinion pieces, and letters to editors. The topic guide was reviewed by the research team and other colleagues with expertise in AMR and infectious diseases. Minor revisions were made according to their feedback. The interview guide was subsequently translated into Bengali. The first author compared the back-translated version with the original version to ensure that a uniformity of meaning between the two languages was achieved. The guide was then piloted with four participants by the first author; these data were not included in the analysis. In keeping with the iterative approach of qualitative research, new questions and areas of inquiry were included as the interviews progressed.

Each interview commenced with introductory questions regarding their current health status, history of chronic disease and COVID-19 infection, understanding COVID-19 infection, health seeking behavior, understanding on antibiotic use, motivators of antibiotic use, and sources of advice. Appropriate probing questions were asked followed by asking broader issues. Each interview lasted 35 to 80 minutes, and were audio-recorded with participants' consent. The recordings were then transcribed verbatim in Bengali and anonymized. The first and

second authors checked all transcripts against the audio recordings for accuracy and completeness.

## Approach to analysis

An inductive, grounded theory approach [52] was followed to capture the themes from the interviewee's explanations of consuming antibiotics in relation to their experience of COVID-19. The interviews were transcribed verbatim. A pseudonym was inserted in each of the anonymized transcripts, which were imported into MAXQDA Standard (2020, VERBI Software, Berlin, Germany) for data analysis. We first listened to the audio recordings and read each transcript to generate initial codes. We then used an inductive coding to analyze the materials. A primary list of codes was developed (by first and second author) and additional inductive codes were added as new insights emerged from the interviews. These codes were then organized into themes and sub-themes for the preparation of research article manuscripts.

## Ethical considerations

Ethical approval of this study was obtained from Chattogram Veterinary and Animal Science University, Chattogram, Bangladesh (CVASU) Research Ethics Committee [approval number CVASU/Dir (R&E)/EC/2020/169(3)]. Prior to taking part in the interviews, participants were informed about the objectives of the research in order to make a well-informed decision as to whether or not they would like to participate. Verbal consent was obtained from the participants and it was recorded in a digitized audio recording prior to starting the guided discussions. Since the study conducted amid the social distancing, and other restrictions imposed to control the spread of the virus, the interviews were conducted over telephone. Therefore, obtaining a written consent was not feasible. The interviews were one-to-one in nature. However, the participants were given opportunity to withdraw from the study at any point, and were informed they that did not have to answer any questions or statements that made them uncomfortable. Identifiable information of the participants was removed during data analysis, and pseudonyms have been used in data reporting. Similarly, the brand names of the antibiotics were also anonymized (with XXX) as it might impact on general public and pharmaceutical companies negatively or positively.

## Results

### Characteristics of the study participants

Participant characteristics are outlined in Table 1. An equal number of interviews were conducted with female and male participants, and most (n = 31) participants were aged 24–40 years. The majority of participants had completed a university degree and most were involved with different public and private agency services. Most of the participants did not have any long-term health conditions. However, more than half of the participants had taken antibiotics in the past year.

### Thematic results

We identified five main themes in the interview accounts of using antibiotics during the COVID-19 pandemic: the attractions of big medicine; managing anxiety about COVID-19; social repercussions of symptoms, diagnosis and isolation; health systems and; sources of advice. Table 2 summarizes the major themes and sub-themes identified during data analysis.

**Table 1. Characteristics of the study participants.**

| Characteristics of the participants | | Number (%) |
|---|---|---|
| Gender | Male | 20 (50) |
| | Female | 20 (50) |
| Age | 24–30 | 15 (37.5) |
| | 31–40 | 16 (40) |
| | 41–50 | 9 (22.5) |
| Education | MA/MSc/MSS | 19 (47.5) |
| | BA/BSc/BSS | 14 (35) |
| | 12<sup>th</sup> grade | 3 (7.5) |
| | 10<sup>th</sup> grade | 4 (10) |
| Occupation | Government service | 12 (30) |
| | Home maker | 11 (27.5) |
| | Private service | 6 (15) |
| | Student | 5 (12.5) |
| | Business | 5 (12.5) |
| | Priest | 1 (2.5) |
| History of chronic disease | Asthma | 4 (10) |
| | Diabetes | 3 (7.5) |
| | Cancer | 1 (2.5) |
| | No Chronic disease | 32 (80) |
| Marital status | Married | 27 (67.5) |
| | Single | 8 (20) |
| | Separated | 1 (2.5) |
| | Not mentioned | 4 (10) |
| Average monthly income (USD) | 60–235 | 6 (15) |
| | 236–470 | 12 (30) |
| | 471–705 | 9 (22.5) |
| | 706> | 9 (22.5) |
| | Not mentioned | 4 (10) |
| Number of times antibiotics taken in last one year | Not taken | 19 (47.5) |
| | One time | 8 (20) |
| | 2–4 times | 10 (25) |
| | Could not recall | 3 (7.5) |

**"Big" medicine.** Participants spoke of antibiotics as powerful drugs that accelerate recovery. They understood antibiotics as the ultimate solution to infection. In this example, Rahman (male, aged 26, who had COVID-19 symptoms) spoke of antibiotics as the 'big boss':

R: Because of cold and cough, I can barely sleep at night. At that time, the doctor prescribed me antibiotic. So that I come to a conclusion that when we do not get cured by taking general medicine to cure our disease, then doctor prescribe us antibiotic. Actually, antibiotic is the big boss.

I: What do you mean by big boss, can you please explain that to me?

R: As I have said, when any medicine does not work, antibiotic works at that time. When doctors find that after giving the medication still the condition of the patient is not improving, then they [doctors] depend on antibiotic. After giving antibiotics, the patient gets recovered. I have seen the same thing in my case too. I was not cured after taking many

**Table 2. Thematic results at a glance.**

| Theme | Sub-theme | Explanation |
|---|---|---|
| **1. Big medicine** | 1.1 Antibiotics are powerful medicine. | Antibiotics are very powerful medicine. The higher the power is, the better they work, and fasten recovery. When the normal medicines do not work, antibiotics are given as final solution. |
| | 1.2 Immunity | Immunity is important to fight against coronavirus. Antibiotics help to improve the immunity. |
| | 1.3 Secondary infection | COVID-19 may cause secondary infection on respiratory tract system. Using antibiotics can prevent that perceived infection. |
| | 1.4 Pneumonia | Patients with COVID-19 may suffer from pneumonia. Taking antibiotics can prevent pneumonia. |
| **2. Managing emotions** | 2.1 Fear | Fear of death and getting infected motivate people to use antibiotics. |
| | 2.2 Relief from mental stress | Having COVID-19 like symptoms causes mental disturbance, anxiety and loss of mental strength. Antibiotics help to gain confidence by quick recovery of the symptoms. |
| | 2.3 Uncertainty of COVID-19 treatment | Uncertainty due to the unavailability of proven therapeutics and vaccine to treat or prevent COVID-19. Hence, antibiotics is the only hope to cure fever, cough and throat itching. |
| **3. Social repercussion of symptoms, diagnosis and isolation** | | Patients with COVID-19 and COVID-19 like symptoms are being stigmatized. It is better use antibiotics rather being stigmatized socially. |
| **4. Health system** | 4.1 Inaccessible health care services. | It is better to consume antibiotics to reduce the symptoms as it is difficult to access to healthcare services from public and private health facilities amid the pandemic. |
| | 4.2 Difficulty in access to COVID-19 testing. | Taking of antibiotic partly solved the problem of deep uncertainty about one's coronavirus antibody status. |
| | 4.3 Pluralistic health services. | Due to the biomedical ambiguity, people prefer to seek health services from other sources and consumed antibiotics by them. |
| **5. Source of advice** | 5.1 Social capital. | Getting advice from the recovered patients and other social network was acceptable as no proven therapeutic was available. |
| | 5.2 Social media | Social media discussions on the (in)effectiveness of taking antibiotics was an influential driver to take antibiotics. A number of respondents reported influenced of social media post to consume antibiotics. |
| | 5.3 Medicine shop. | People frequently consumed antibiotics based on pharmacy workers' and owners' suggestions. |
| | 5.4 Quack and non-human physicians | Many respondents consumed antibiotics based on the suggestions from quacks and non-human physicians like vets. |

medicines. At last, I was suggested with antibiotic, I took antibiotic and get cured. That is why I called it 'big boss'.

Rahman used the metaphor of "big boss" to capture what he saw as the healing powers of antibiotics. The fragment also makes reference to previous experiences with treatments to back up this view. These notions of antibiotic powers recurred in the interviews and were recognized as one reason why people sought them in the time of pandemic threat. In the following example, Taslima (female, aged 32, COVID-19 patient) noted that antibiotics are not used to treat COVID-19 but explained that people may seek them because they are the 'biggest' medicine:

In my knowledge, antibiotics should not be used for COVID-19. But recently, people are using XXX [brand name of antibiotics] along with other medicines. Now the question is why? Because, you know, it is popularly known that antibiotics are the biggest medicine

and very powerful. . . When they [people] suffer from fever, throat ache, cold, pain or diarrhea people take antibiotics. People know that, antibiotics are useful.

Taslima took the observing position in her narrative and reported on the behavior of others, a common storytelling approach for topics that may cast doubt on the protagonist's knowledge and judgement. Taslima's narrator position is an important clue that some awareness exists that the unjustified use of antibiotics is not sanctioned by experts. The reputation of antibiotics as big medicine was echoed in talk about speedy recovery, another source of value for consumers. But some also linked this power with their COVID-19 outcomes; for example, Rafique (male, aged 27, COVID-19) explained:

I can tell you that it took six days to become negative from the day I got positive with COVID-19. I started using antibiotics on that day when I got my COVID-19 positive report. I also used other normal medicines along with azithromycin. In that case, using antibiotic helps me to recover fast. On the other hand, it took 15 days to get recovered from COVID-19 for my parents. They did not take antibiotics. . . I cannot be certain about this but this is my observation.

In this example, Rafique compared his own quick recovery with antibiotics with his parents' experience of longer period of COVID-19 symptoms. This is another example of experience being used to justify the use of antibiotics, despite its inaccuracy. This interpretation suggests that the reputation of antibiotics is promoted through lack of knowledge about viruses, bacteria and antibiotics. Participants with COVID-19 like symptoms also frequently reported similar explanations. 'Antibiotics are very powerful medicine' and 'when other medicines do not work well, antibiotics are used' were frequently mentioned explanations for taking antibiotics.

Other participants interpreted symptoms as possible COVID-19 infection. These included fever, cough, sore throat, itchy throat and body ache. When participants noticed these symptoms, they started thinking that they may have the virus. According to Masum (male, aged 50, COVID-19 like symptoms):

Just before the nationwide lockdown, I was travelling to outside of Dhaka for an official purpose. On my return to Dhaka, I was suffering from throat ache and felt itching on my throat. Then I checked temperature and found 100-degree Celsius. I thought it was a seasonal flu. I took normal paracetamol along with a syrup for cough. But my condition was not improving. Then I got confused. As the COVID situation was getting worse day by day then, I assumed to have the virus, because these symptoms are common for coronavirus. I did not want to take risk with these symptoms. Then I took antibiotics and recovered from fever and throat ache and itching.

A set of explanations put forward the notion that antibiotics keep the immune system active and strong. Rahima (female, aged 26 and COVID-19) stated,

COVID-19 is not a bacterial disease; rather it is viral. Antibiotics do not work for viral disease but help to improve immunity. For this reason, I used antibiotics. You know, coronavirus impacts on whole body. Body cannot fight against the virus but if your immune system is strong then your body can fight against further damage caused by COVID-19.

Rahima justified the use of antibiotics to make the body fit to fight against the potential damage caused by SARS-CoV-2.

A few participants reported that antibiotics were given to prevent secondary infection and in case of pneumonia. The WHO advises that antibiotics are only be given to confirmed cases of secondary infection or pneumonia. In contrast, our participants' explanations demonstrate people are taking antibiotics outside these guidelines. Sohel (41, patient with COVID-19) explained,

> The aged people and people with previous disease [co-morbidity] are at higher risk to COVID. Their condition becomes severe on 7th or 8th day of infection. As COVID is a lung disease, there is huge chance to spread the virus into the whole respiratory system. As a result, a COVID patient may suffer from severe inhalation problem, which may cause further complications, even death. I did not want to have such complication. To prevent this complication, I took antibiotics.

Sohel's explanation was typical of our interviewees. Antibiotics are justified as a means to avoid the progression of illness to severe debility and death. These explanations show that antibiotics' reputations as powerful drugs lead people to seek them out after diagnosis or when they experience symptoms.

**Managing emotions.** Participants reported considerable anxiety and fear connected with the pandemic in general, their diagnosis and symptoms, and linked these emotions with antibiotics. They referred to fear of death, fear of getting infected with COVID-19, the extensive uncertainties regarding COVID-19 treatment, intensely negative emotions, and stress relation to social stigma. People who were tested positive for COVID-19 and those with COVID-19-like symptoms referred to these concerns, but with different nuances depending on their diagnosis. Sattar (male, aged 44, COVID-19 patient) spoke of antibiotics as a way to manage his anxiety,

> Currently, people are getting anxious due to COVID-19. Nobody knows what to do if they got infected by the virus. I am also thinking in a similar manner. Therefore, I took antibiotics thinking that it would help me to reduce my fear. In my opinion, using antibiotics can help me to get rid of current worries.

The 'big' medicine reputation of antibiotics was therefore a source of some emotional relief, helping to explain why they may be sought out by patients despite not being clinically useful. One issue here is that from the patient's point of view, antibiotics may appear to be effective, when an infection resolves of its own accord. For example, Nazma (female, aged 30, with COVID-like symptoms) spoke of her symptoms subsiding after she took antibiotics, which was linked with a more general concern to reduce her anxiety,

> When I was suffering from fever and throat ache, I was afraid to be positive with COVID-19. Then I became aware by myself that I should not be sick. My primary symptoms told me that maybe I was positive. That was a kind of mental disturbance. But after taking antibiotics, I was confident about that I am not positive as my throat ache went away. . . .. I was very tense, if I were positive then my family members will also be positive and I will be responsible for that. That feeling disturbed me a lot. For all these tensions and mental satisfaction, I took antibiotics.

Like Nazma suggested, having a diagnosed patient in the family home motivated some participants to take antibiotics. When a family member is diagnosed, other members are at risk, which appeared to generate some tension. In this situation, using antibiotics was reported to

be a coping mechanism for reducing risk and therefore anxiety. Ruma (female, aged 40, with COVID-like symptoms), whose husband had been diagnosed with COVID-19, explained that situation,

> My husband was tested positive with COVID. As I took care of him, my husband's doctor also suspected that I may get the virus. I did not have all the symptoms but was suffering from weakness. But as my husband was positive, so I was guessing I might have the virus as well. For that reason, I also took his medicines. My husband brought antibiotics and other medicines. So I started taking all of his medicines.

Ruma's comment also suggest that she 'self-diagnosed' by assessing her symptoms and proximity to someone who had been diagnosed. She also spoke of how antibiotics and other pharmaceuticals were shared between family members. Antibiotics are seen as powerful medicines and for that reason have psychological properties, particularly in the time of a life-threatening pandemic for which effective vaccines and treatments are not yet available.

**Social repercussions of symptoms, diagnosis and isolation.** A number of explanations from the people with COVID-19 like symptoms were linked with social stigma. As part of control and containment mechanisms, people with COVID-19 have to stay isolated, maintain quarantine, and are not permitted to go to communal places. Misunderstanding isolation and quarantine was a source of social rejection and vilification, which meant that the illness had social costs that had to be avoided. Nasrin (female, aged 29, with COVID-like symptoms) explained the situation in following way:

> The COVID patients are being isolated socially. We have also seen COVID patients are being harassed socially. Their families are suffering a lot by the neighbors. Their electricity and water connection cut, they cannot even collect their daily needs. Children abandoned their aged parents. I have seen such cases in television and newspaper. Considering these, I was aware of these and I was strict to be healthy. To avoid all these complications and problems, I took antibiotics to show that I am not sick.

Nasrin's interview showed a key social driver of antibiotic use in the time of pandemic are the social sanctions on people with the virus. We can surmise too that these dynamics may reduce testing for COVID-19 and accentuate self-diagnosis and self-treatment. Morshed (male, aged 30, with COVID-like symptoms) provided a picture of some distress due to an uncontrollable cough and its impact in social settings,

> I was suffering from cough and fever. I discussed a local quack [unprofessional prescriber] near to my home. I told her that I am coughing a lot when I travel and in the office. When you have cough, you cannot control it. Now people are more concern about coughing and sneezing in the public. I tried to control it but could not. People would have looked at me differently. . . so I did not want to take that risk. I shared my problem with that doctor and asked what should I do? Then she suggested me to take antibiotics to prevent the COVID and get cured from coughing. Now I am feeling good, by the grace of Allah.

As in many other examples in the interviews, antibiotics solved symptoms, negative emotions and threats to social standing. The effects of antibiotics were almost magical since they erased these problems, even though they were not impacting on the virus or the infection-related symptoms which may have resolved of its own accord. We can see, then, how

antibiotics gain their reputations as powerful medicines as they indeed have many powers in the eyes of people who use them.

**Inaccessible COVID-19 testing and healthcare services.**   Another set of descriptions corresponded with the difficulty accessing COVID-19 testing and health care services. While both participant groups frequently mentioned difficulty in getting access to treatment from government hospitals, those with COVID-19 like symptoms were more likely to consider antibiotics as an alternative to the COVID-19 test. The following explanation of this situation came from Sharmin (female, aged 36 years), a patient with COVID-19 like symptoms,

> Before taking antibiotics, I was suffering from fever and throat ache. I did not do my COVID test. It is difficult to test in Chittagong, as the number of laboratories for COVID tests is very limited. People cannot do test if they want to do so. It takes 3 or more days to get seen for a test and then it requires another 2 or 3 days to get reports. So, for me, doing test is very difficult and a hassle. I did not bother to do the test. At that time, I thought it would be better to use antibiotics, so that I can recover myself from fever and throat ache.

Like many other patients with COVID-19 symptoms, Sharmin was motivated to use antibiotics as she was not able to do the test. In this context, using antibiotics stood in for the COVID-19 test or bypassed it. In this way, the antibiotic partly solved the problem of deep uncertainty about one's coronavirus status. However, access to general and COVID-19 specific health services remained difficult for many of our research participants. Consequently, people had to rely on home-based care services they organize themselves. As Zahir (male, aged 28 years, with COVID-like symptoms) stated,

> When I was suffering from fever and cough, I was thinking about my wife. Last month when my wife was suffering from another disease, I took her to the hospital. I saw people suffering because they could not get general treatments. The doctors and nurses were feeling worried if they would get infected by the patients. The hospitals even were not admitting new patients. The hospital where I took my wife made the COVID test mandatory, otherwise they would not admit patients. Now everyone is not able to do the test. So, people had to go home [without taking treatment]. Seeing these situations, I thought many people may die at home, and not from COVID. I was thinking, if I suffer from any other diseases, maybe I cannot get proper treatment. So, to be healthy, I took antibiotics.

**Sources of advice.**   Another set of explanations for antibiotic use was linked with pluralistic health care and healing systems. The thematic analysis revealed that participants relied on the experience and advice of a wide range of variously qualified people from their social worlds, including recovered patients, social networks, professional doctors, veterinarians, and 'quacks'. The following explanation showed how Rokhsana (female, aged 34), herself a recovered COVID-19 patient, was motivated to use antibiotics through the example of a friend who had recovered,

> One of my acquaintances tested positive with coronavirus. She was suffering from fever, cough and throat ache. When her condition was not improved, she took antibiotics. Her condition improved rapidly after taking antibiotics. When I was tested positive, I called her and she suggested me to take antibiotics. She suggested me to take one tablet in the morning and another at night. Three days later (of taking antibiotics) my fever and throat ache

went away. I took antibiotics based on her suggestion, as she recovered from COVID-19, I thought she could be an ideal source how to manage.

A large number of participants reported that they had used antibiotics based on the suggestion from medical practitioners. Although their doctors did not explain why antibiotics were to be used, they cited such advice in their justifications. The following explanation from Rubel (male, aged 42, with COVID-19) depicted this situation,

I cannot tell you detail why I used antibiotics. When I was tested positive, a doctor from a government hospital called me. He asked me about my health condition. I told him in detail. Then he suggested few medicines. While buying those medicines I came to know there was a medicine of antibiotic group. I used those medicine as the doctor suggested.

Excessive prescription of antibiotics has been considered as a key driver of AMR, so it is unknown why doctors are prescribing antibiotics to home-based patients for treating COVID-19. In general, high antibiotic prescription by medical doctors has been reported in Bangladesh [23]. Rubel's account indicates that inconsistent application of antibiotic use guidance leads to some confusion among consumers.

A number of explanations indicated that people in participants' social networks influenced them to take antibiotics when they found themselves to have COVID-19 or symptoms. Taukir (male, aged 32), a patient with COVID-19, stated,

I cannot say in detail about the benefits of using antibiotics... I heard from my personal networks; antibiotics are very helpful for COVID. Hearing from my personal network, I also felt that, using antibiotics would improve my condition. Nothing else, you know people do not take medicine for fun. Usually, I avoid using medicine for simple illness. As there's no specific treatment for COVID-19, I took antibiotics and found them effective and that's why using antibiotics is becoming popular.

In addition, some research participants mentioned that their antibiotic use decisions were motivated by social media, including those with COVID-19 and COVID-19 like symptoms. For example, Hanufa (female, aged 31, with COVID-19) used antibiotics based on social media post. According to her,

I got know from the social media about taking antibiotics. I saw many Facebook posts discussing how antibiotics are useful for COVID. I can talk about one post of my close friend where she was saying antibiotics will help to you feel comfortable, at-least for throat ache. My husband [was also a COVID patient] also saw other posts. Then me and my husband started using.

During the COVID-19 crisis, social media appear to play a significant role in disseminating (mis)information about how to manage symptoms and use antibiotics. In the context of fragmented or non-existent access to testing and diagnosis services in Bangladesh, social media messages may take a central role in the self-care decisions of people affected by the pandemic.

The interviews also revealed that a smaller number of participants took antibiotics based on suggestions from veterinary doctors. Although the vets were not supposed to suggest antibiotics for use in human health, during the pandemic crisis they have been considered to be sources of expert advice. For example, Jahanara (female, aged 29, with COVID-19) explained,

> I took antibiotics based on my neighbor's suggestion. He is a veterinary doctor. He completed DVM from CVASU [Chittagong Veterinary and Animal Science University] . . . as he has expertise on animal medicine, I thought his suggestion for taking antibiotic is appropriate. Like humans, animals also need antibiotics.

Similar explanations were observed from the five participants who used antibiotics based on advice from 'quacks', who are non-medical persons who work as assistants to professional doctors. These participants explained that quacks are trusted because they are perceived to be up-to-date with current recommendations for the treatment of COVID-19.

## Discussion

This exploratory study has documented the key drivers for using antibiotics among people in Bangladesh in a time when the world is facing an unpreceded health crisis due to COVID-19. The study has revealed that popular norms regarding the powers of antibiotics, psychosocial crises due to the pandemic, humoral understandings of health, social repercussions of symptoms, diagnosis and isolation, inaccessible COVID-19 testing and multiple sources of advice and expert knowledge all played important roles.

Antibiotics were understood to be effective when other medicines do not reduce the symptoms of COVID-19, including fever, throat itching, body ache, diarrhea, cold cough and smelling sensation. Humoral understandings [31] such as 'antibiotics are powerful', 'final solution' and 'antibiotics are big boss' significantly motivated patients and undiagnosed people to use antibiotics when they experienced COVID-19 like symptoms. This is an extension of people's pre-pandemic health management practices which highlight understandings of antibiotic and antibiotic use that differ from biomedical perspectives according to socio-economic status and cultural context [31, 43]. Moreover, the explanations offered in our research, underline how people make sense of their health according to their social, cultural and prior experiences [53–55], perhaps most keenly in the time of pandemic. As is evident in this study, participants prioritized the symptoms of COVID-19 above biomedical knowledge regarding the prudent use of antibiotics, a finding which is aligned with an anthropological study in India [56].

While discussing their motivations for using antibiotics during COVID-19, study participants indicated that the cultural meaning of 'rational use' from consumers' perspective is different from the top-down, expert-led approach [42, 43]. Respondents' explanations demonstrated how these decisions were considered responses to the psychological and social repercussions of COVID-19 infection and symptoms, inaccessible COVID-19 testing, social media influences and efforts to avoid secondary co-infection. Each sub-theme showed that taking antibiotics helped individuals to cope with the physical and psychosocial stressors of life during the pandemic. Concern about the mental health challenges among the general population, COVID-19-infected patients, close contacts, elderly, children and health professionals has emerged as one of the pandemic's 'hidden' impacts [57]. In the pre-pandemic era, a number of studies reported that people with greater psychological distress and lower self-efficacy report high levels of intention to use antibiotics [58–60]. Mental health strains caused by confusion and uncertainty around COVID-19 treatment and delay in finding a proven vaccine may also contribute to the use of antibiotics amid this pandemic [12, 61].

While antibiotics are popularly known as 'big medicines' that work to cure infections [20, 34, 62], the current study has also revealed that antibiotics are used to prevent the potential of secondary infection, such as pneumonia, caused by coronavirus infection. The explanations we examined indicated that participants understand the potential for secondary infections and use antibiotics of their own accord or based on the advice of various 'experts', despite the

'inappropriateness' of such practice [43]. This prophylactic use of antibiotics aligns with Nichter's (2001) work which showed that sex workers in the Philippines routinely used antibiotics before and after sex to prevent potential bacterial infection, and the use of antiviral treatments to prevent HIV infection [63]. Antibiotics are widely used in clinical settings with(out) clinical confirmation of lung infection [64], which may be partially responsible for the antibiotic prophylaxis among patients in home settings.

The social repercussions of COVID-19 symptoms, diagnosis and isolation were additional drivers for choosing antibiotics. Study participants explained that people diagnosed with COVID-19 and COVID-19 like symptoms were socially stigmatized, leading patients to experience neglect, marginalization, and ostracization, as has been observed for other pandemics and outbreaks [65–69] which amplify hate, discrimination, chaos, fear, and violence. As antibiotics are considered 'magic drugs' that can hasten recovery, the consumption of antibiotics helps individuals to avoid these daunting social repercussions of COVID-19. In addition to the reputation of antibiotics and psychosocial considerations, our analysis showed how antibiotic use is embedded in the structures of the health system. Uneven access to testing and weak public health systems infrastructure, in general, are considered major factors for the spread of COVID-19 in LMICs [18, 19]. Our respondents showed how these factors led some to consume antibiotics, with some noting that antibiotics provided some assurance for the mitigation of infection in the absence of a COVID-19 test. This insight is important because it points to the health system drivers of antibiotic consumption and therefore AMR.

In addition, recommendations from sources within the pluralistic health system were also linked with accounts of taking antibiotics. Literature on the pre-pandemic era shows that inappropriate prescription practices by physicians was one of the major drivers of antibiotic use [23, 25]. This driver was also discussed by the participants of our study. A recent study in Zimbabwe showed that antibiotics are seen by the frontline prescribers as a 'big gun', 'thick bleach' and 'cover' for novel diseases [70]. As COVID-19 is a relatively new disease, this 'big medicine' reputation of antibiotics might influence inappropriate practices of antibiotics by the prescribers. In addition, our study participants relied on alternative medical services, in particular, popular sector (e.g., COVID-19 survivor), non-medical providers (quacks), and veterinary doctors. This reliance on experts outside the human health system echoes anthropological literature on the significance of multiple healing systems in LMIC contexts [43, 56]. For most of our study participants, suggestions from recovered patients within their social network were important for their decisions about treatment, medicine and care for COVID-19. Social media are commonly seen as important methods for raising AMR awareness [71, 72]. However, our study participants were influenced by social media, particularly Facebook, to use antibiotics for COVID-19, indicating an urgent need to provide easily accessible advice online to moderate these influences.

This explanatory research has some limitations that apply to the findings. First, it was conducted in two major urban areas with highly dense populations that may not be representative of the whole of Bangladesh, particularly rural areas. However, the current study covered the most affected areas in terms of number of COVID-19 cases. Second, study participants were mostly highly educated and of similar economic backgrounds, which may reflect the recruitment strategies (network sampling and Facebook). Nevertheless, it is worth understanding 'higher educated' people's perspective on antibiotic use during COVID-19 as it is assumed people with higher education group perform better in terms appropriate use of antibiotics. Finally, due to the nature of the research approach, the study was conducted among people with COVID-19 and COVID-19 like symptoms who received care in home settings, while hospital-based patients were excluded. Our findings provide insights for further investigations of COVID-19 related antibiotic use in hospital settings.

## Conclusion and recommendations

AMR researchers, advocates and public health specialists have been concerned about the observed increased use of antibiotics during the COVID-19 pandemic [1, 73, 74]. Our study offers some important insights into the mechanisms and drivers of the use of antibiotics in Bangladesh during the pandemic. We were able to document the key social, cultural and psychological properties of antibiotics that help to explain why they are sought out in a time of health crisis, even amongst those who recognize that antibiotics cannot treat viral infections. We also shed light on the health systems and social media drivers of antibiotic use, in time of crisis. The study demonstrates that effective interventions for AMR awareness will need to do more than simply provide information. They will need to assist members of the general public to take action to safeguard their health while also moderating the use of antibiotics. Messaging will need to acknowledge the reputation of antibiotics and their socio-pharmacological value, and help people to understand how antibiotics can be used to best effect and to reduce AMR. Our research also indicates that national strategy for AMR reduction will also have to address health system drivers of AMR and social media influences. The lessons that the pandemic has provided about the drivers of antibiotic use are important in the short term as public health systems across the world seek out ways through the health crisis. Our analysis is also valuable in the longer term because, without strong action, the AMR challenge will continue and deepen, in part due to the impact of COVID-19. Moreover, the advent of the pandemic and its impact on AMR are important considerations for the public health response to future pandemics. The study strongly recommends further research on clinical usage of antibiotics, physicians' theory of practice while prescribing antibiotics, the role of pharmaceuticalisation, and multidisciplinary political and policy analyses of antimicrobial use in the pandemic context in Bangladesh and beyond.

## Supporting information

**S1 File. Interview guide.**
(DOCX)

**S2 File. Supplementary file.**
(DOCX)

## Acknowledgments

We would like to thank the research participants who gave their valuable time and took participate in the study voluntarily. We also thank to Chattogram Veterinary and Animal Science University for giving ethical approval of this study. The authors also thank to the RAs who conducted all interviews. The authors acknowledge with particular gratitude the anonymous reviewers who offered detailed and helpful comments on the manuscript.

## Author Contributions

**Conceptualization:** Abul Kalam, Shahanaj Shano, Mohammad Mahmudul Hassan.

**Formal analysis:** Abul Kalam, Shahanaj Shano, Mark Davis.

**Funding acquisition:** Mohammad Mahmudul Hassan.

**Investigation:** Mohammad Asif Khan.

**Methodology:** Abul Kalam.

**Project administration:** Abul Kalam, Shahanaj Shano, Mohammad Asif Khan, Ariful Islam, Mohammad Mahmudul Hassan.

**Software:** Abul Kalam.

**Supervision:** Mohammad Mahmudul Hassan, Mark Davis.

**Validation:** Shahanaj Shano, Mohammad Asif Khan, Ariful Islam, Narelle Warren, Mark Davis.

**Writing – original draft:** Abul Kalam, Shahanaj Shano.

**Writing – review & editing:** Abul Kalam, Shahanaj Shano, Mohammad Asif Khan, Ariful Islam, Narelle Warren, Mohammad Mahmudul Hassan, Mark Davis.

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
