## [Decision Letter · Decision Letter 0]

29 Sep 2021

PONE-D-21-18370Understanding Antibiotic Consumption Amongst Individuals with COVID-19 or Symptoms: Implications for Antimicrobial Resistance in BangladeshPLOS ONE

Dear Dr. Kalam,

Thank you for submitting your manuscript to PLOS ONE. After careful consideration, we feel that it has merit but does not fully meet PLOS ONE’s publication criteria as it currently stands. Therefore, we invite you to submit a revised version of the manuscript that addresses the points raised during the review process.

During the revision process while you address the reviewers comments, please revise the format of your manuscript. It needs to follow Plos One guidelines, so please pay attention to the guidelines.Also, the names listed on the manuscript could create conflict and a potential breach in patient confidentiality, please use codes instead names.

We look forward to receiving your revised manuscript.

Kind regards,

Monica Cartelle Gestal, PhD

Academic Editor

PLOS ONE

a) Did participants provide their written or verbal informed consent to participate in this study?

3. When reporting the results of qualitative research, we suggest consulting the COREQ guidelines  or other relevant checklists listed by the Equator Network, such as the SRQR, to ensure complete reporting (http://journals.plos.org/plosone/s/submission-guidelines#loc-qualitative-research). Moreover, please provide the interview guide used as a Supplementary File

“We also grateful to Bangladesh Bureau of Educational Information and Statistics for providing partial funding (ID#SD2019967) to this study.”

“This work is partially funded by Bangladesh Bureau of Educational Information and Statistics (BANBEIS), ID number SD2019967. Mohammad Mahmudul Hassan was supported by BANBEIS.”

6. We note that you have indicated that data from this study are available upon request. PLOS only allows data to be available upon request if there are legal or ethical restrictions on sharing data publicly. For more information on unacceptable data access restrictions, please see http://journals.plos.org/plosone/s/data-availability#loc-unacceptable-data-access-restrictions.

7. Your ethics statement should only appear in the Methods section of your manuscript. If your ethics statement is written in any section besides the Methods, please delete it from any other section.

Additional Editor Comments (if provided):

Reviewers' comments:

Reviewer's Responses to Questions

**Comments to the Author**

1. Is the manuscript technically sound, and do the data support the conclusions?

Reviewer #1: Yes

Reviewer #2: Yes

Reviewer #3: Partly

2. Has the statistical analysis been performed appropriately and rigorously? 

Reviewer #1: N/A

Reviewer #2: N/A

Reviewer #3: I Don't Know

3. Have the authors made all data underlying the findings in their manuscript fully available?

Reviewer #1: No

Reviewer #2: Yes

Reviewer #3: No

4. Is the manuscript presented in an intelligible fashion and written in standard English?

Reviewer #1: Yes

Reviewer #2: Yes

Reviewer #3: No

5. Review Comments to the Author

Reviewer #1: Overall a fascinating manuscript where the study is well done and the topic is extremely relevant to the current twin pandemics of AMR and COVID.

Major critique:

1. Authors mention that antibiotic use has gone up in during COVID. However, that is not true globally as many outpatient facilities in the US have actually seen a decrease in antibiotic prescription (in the US as eg. https://academic.oup.com/cid/advance-article/doi/10.1093/cid/ciaa1896/6054971). So would recommend stating that and then commenting that unfortunately the overuse of antibiotics is worse in developing countries that already have a high burden of covid and a future increased burden of AMR would cripple the healthcare systems further.

2. The authors only make a cursory mention of inappropriate prescription practices by physicians. Since this was bought up by the participants as well, I would recommend discussing this in more detail using pre-pandemic literature as well.

3. How did the authors confirm that antibiotics were indeed being used by their participants? For eg. did they ask them for specific names of the prescriptions? Many times people use the term 'antibiotic' generally for any medicine where even ibuprofen is considered an antibiotic. Also would recommend mentioning which antibiotics the participants identified either using of getting prescribed.

4. The authors have used first names of participants. Please use First and Last initials to maintain confidentiality. Also would recommend adding age and gender to the initials. Eg. state MA (26yrs Female) said 'xxxx....'.

Minor critique:

Page 4 (line 89-91): sentence should be rephrased to increase clarity

Page 5 paragraph titled "COVID-19 situation and response in Bangladesh' seems oddly placed. Consider moving it prior to the previous paragraph (lines 96-104)

Page 6 line 134: Is that the total number of COVID cases in Bangladesh at the time? Please clarify. Also 'Residents both in cities' is a typo.

Reviewer #2: This is a very interesting and well written qualitative study of antibiotic use among people with COVID-19 positive test or symptoms in Bangladesh. This was a fascinating read and I very much appreciate the authors’ work. I have a few comments mostly meant for clarification.

1. Introduction – It would be helpful to provide a citation for the statements in the first two sentences (lines 45-48).

2. Introduction, Line 55 – typo in “…related mortality” (rather than morality)

3. Study setting, Line 133 – might want to clarify that the 70.6% and 14.1% figures are % shares of total number of cases, not a population prevalence of COVID (e.g., 70% of the population of Dhaka does not have COVID at any given time).

4. Study participants – was there a timeframe during which participants had to have tested positive for COVID and/or were symptomatic? Rather than listing gender/age as an inclusion criteria, I would note separately that purposeful sampling was used (or something, whatever is accurate) to ensure an even distribution of gender and age.

5. The inclusion criteria state that taking antibiotics in the last 12 months was an inclusion criterion, but the table 1 characteristics state that 19 participants hadn’t taken antibiotics in the last year. Was past-year antibiotic use a criterion for inclusion in this study?

6. I think Table 1 and associated results text should be in the results of the paper, not the methods. Additionally, it would be helpful to revise Table 1 into a more standard format with one row per characteristic (rather than two columns with characteristics), include both Ns and % of the population, etc. In general, I would use standard section headings (Introduction, Methods, Results, Discussion) and sub-headings within each of those broader categories (e.g., for Study Participants).

7. Were there participants in the sample who had not taken antibiotics for COVID? Were there any themes or results emerging from these interviews?

Reviewer #3: - The title of the manuscript should be edited to better reflect the outcomes of this study. What do you mean by "Symptoms" in the title? It should be COVID-19-like symptoms...

- The English writing of the manuscript needs proofreading.

- How the authors diagnosed COVID-19 infection? This should be clearly explained.

- What types of antibiotics were consumed by patients? and for how long?

- Were there any secondary bacterial infections? if yes, which infections?

- The format of Tables needs edition.

- The structure of manuscript is somehow strange. I do not know it is in the format of journal. For example, findings instead of Results!!!

- The limitations of the study should be mentioned in the discussion.

- The following article fully explained the AMR situation during COVID-19 pandemic. Cite it in the Introduction or Discussion part.

doi: 10.3389/fmicb.2020.590683

"Antimicrobial Resistance as a Hidden Menace Lurking Behind the COVID-19 Outbreak: The Global Impacts of Too Much Hygiene on AMR"

Front Microbiol. 2020; 11: 590683.

6. PLOS authors have the option to publish the peer review history of their article (what does this mean?). If published, this will include your full peer review and any attached files.

Reviewer #1: No

Reviewer #2: No

Reviewer #3: No

---

## [Author Response · Author response to Decision Letter 0]

22 Oct 2021

October 7, 2021

We are pleased to provide a revised manuscript entitled: Understanding the Social Drivers of Antibiotic Use During COVID-19 in Bangladesh: Implications for Reduction of Antimicrobial Resistance. The title has been revised as per suggested by respected reviewer. We thank the reviewers for their overall enthusiasm for the study and their constructive comments, which have allowed us to improve the manuscript significantly. 

We have responded to each of their comments, as detailed below. Our manuscript has even more relevance at a time when the world is experiencing a severe pandemic, and there are lot of anxiety and tensions around antibiotics use. Therefore, studying social drivers of antibiotics use, this study provides insight that will inform health policy across different countries including Bangladesh.

We look forward to the review of our revised manuscript and hope that it is now considered acceptable for publication in PLoS One. 

Sincerely,

Md. Abul Kalam.

Specific responses to academic editor and reviewers:

Comment 1. Please ensure that your manuscript meets PLOS ONE's style requirements, including those for file naming. The PLOS ONE style templates can be found at

Response: Thank you very much. The draft meets the style requirements of the journal. 

Comment 2. Please amend your current ethics statement to address the following concerns: 

a) Did participants provide their written or verbal informed consent to participate in this study? 

Response: Yes. Verbal consent was obtained from all participants. 

Comment b) If consent was verbal, please explain i) why written consent was not obtained, ii) how you documented participant consent, and iii) whether the ethics committees/IRB approved this consent procedure.

Response: Thank you for your concern. We obtained verbal consent in the two stages. At first, the study leads made communication with the potential respondents to explain the study purpose, research objectives and nature of participation. At this stage the study leads got verbal consent and provided a suitable time for the interview. Once a potential participant agreed to participate in the study, the Research Assistants then called them to conduct the interview. Before starting the interview, the research assistants again received informed consent verbally. The verbal consent was audio recorded with permission of the participant before starting the discussion. Since the study was conducted amid strict restrictions due to COVID-19 (such as travel restriction, ban of transportations and physical distancing), it was not possible to obtain written consent. Considering this, the Ethics Committee has agreed on obtaining verbal consent. Please check the process in the Ethical consideration section; line 208-212. 

3. When reporting the results of qualitative research, we suggest consulting the COREQ guidelines or other relevant checklists listed by the Equator Network, such as the SRQR, to ensure complete reporting (http://journals.plos.org/plosone/s/submission-guidelines#loc-qualitative-research ). Moreover, please provide the interview guide used as a Supplementary File. 

Response: We followed COREQ guideline while reporting the results. Please check line number 138-139 in the methodology section. Please find the filled-up checklist form in S2 File. The interview guide has been provided as Supplementary File. Please check S1 File. 

Comment 4. Thank you for stating the following in the Acknowledgments Section of your manuscript:

“We also grateful to Bangladesh Bureau of Educational Information and Statistics for providing partial funding (ID#SD2019967) to this study.”

“This work is partially funded by Bangladesh Bureau of Educational Information and Statistics (BANBEIS), ID number SD2019967. Mohammad Mahmudul Hassan was supported by BANBEIS.”

Response: Thanks for your suggestion. We have removed the funding information from the acknowledgement section. Please check line number 572-573. 

Comment 5. In your Data Availability statement, you have not specified where the minimal data set underlying the results described in your manuscript can be found. PLOS defines a study's minimal data set as the underlying data used to reach the conclusions drawn in the manuscript and any additional data required to replicate the reported study findings in their entirety. All PLOS journals require that the minimal data set be made fully available. For more information about our data policy, please see http://journals.plos.org/plosone/s/data-availability. 

Important: If there are ethical or legal restrictions to sharing your data publicly, please explain these restrictions in detail. Please see our guidelines for more information on what we consider unacceptable restrictions to publicly sharing data: http://journals.plos.org/plosone/s/data-availability#loc-unacceptable-data-access-restrictions . Note that it is not acceptable for the authors to be the sole named individuals responsible for ensuring data access.

Response: Thanks so much for pointing this out. Following COREQ checklist, we have provided minimal data set as the supplementary file 2: S2 File

Comment 6. We note that you have indicated that data from this study are available upon request. PLOS only allows data to be available upon request if there are legal or ethical restrictions on sharing data publicly. For more information on unacceptable data access restrictions, please see http://journals.plos.org/plosone/s/data-availability#loc-unacceptable-data-access-restrictions .

Response: Thanks so much for pointing this out. We have updated the data availability statement. 

7. Your ethics statement should only appear in the Methods section of your manuscript. If your ethics statement is written in any section besides the Methods, please delete it from any other section.

Response: Thanks so much. We have mentioned it only in the methodology section. 

Additional Editor Comments (if provided):

Reviewers' comments:

Reviewer's Responses to Questions

Comments to the Author

1. Is the manuscript technically sound, and do the data support the conclusions?

Reviewer #1: Yes

Reviewer #2: Yes

Reviewer #3: Partly

2. Has the statistical analysis been performed appropriately and rigorously?

Reviewer #1: N/A

Reviewer #2: N/A

Reviewer #3: I Don't Know

3. Have the authors made all data underlying the findings in their manuscript fully available?

Reviewer #1: No

Reviewer #2: Yes

Reviewer #3: No

4. Is the manuscript presented in an intelligible fashion and written in standard English?

Reviewer #1: Yes

Reviewer #2: Yes

Reviewer #3: No

5. Review Comments to the Author

 

Reviewer #1: Overall a fascinating manuscript where the study is well done and the topic is extremely relevant to the current twin pandemics of AMR and COVID.

Response: Thank you so much for your appreciation on the merit our manuscript. Many thanks for your time and efforts that you made to review our paper. We have responded and made necessary changes based on your critical comments, questions and suggestions. 

Major critique:

1. Authors mention that antibiotic use has gone up in during COVID. However, that is not true globally as many outpatient facilities in the US have actually seen a decrease in antibiotic prescription (in the US as eg. https://academic.oup.com/cid/advance-article/doi/10.1093/cid/ciaa1896/6054971 ). So would recommend stating that and then commenting that unfortunately the overuse of antibiotics is worse in developing countries that already have a high burden of covid and a future increased burden of AMR would cripple the healthcare systems further.

Response: Thanks so much for providing this insight with reference. We have now updated the introduction section with this information. Please check line number 48-51. 

Comment 2. The authors only make a cursory mention of inappropriate prescription practices by physicians. Since this was bought up by the participants as well, I would recommend discussing this in more detail using pre-pandemic literature as well.

Response: Thanks so much for you suggestion. We have now elaborated this aspect in the discussion section. Please check line number 521-526. 

Comment 3. How did the authors confirm that antibiotics were indeed being used by their participants? For eg. did they ask them for specific names of the prescriptions? Many times, people use the term 'antibiotic' generally for any medicine where even ibuprofen is considered an antibiotic. Also, would recommend mentioning which antibiotics the participants identified either using of getting prescribed.

Response: We agree with your comment. As we mentioned in the methodology section, the lead researchers had an initial discussion with the potential respondents before enrolling the study. At that time, the potential participants were asked the names of the medicines they consumed. All the participants mentioned the brand names of the antibiotics. One researcher correlated these brand names with type (group) of antibiotics. When any participant failed to mention the name and the brand name did not match with the type of human antibiotics, they were excluded from the study. Please check this explanation in the methodology section, line number 156-160. In regards to the name of antibiotics, as mentioned earlier, participants mentioned the brand name. We are not interested to use these names because, these may impact on the readers or to the specific pharmaceuticals negatively or positively. 

Comment 4. The authors have used first names of participants. Please use First and Last initials to maintain confidentiality. Also would recommend adding age and gender to the initials. Eg. state MA (26yrs Female) said 'xxxx....'.

Response: Thanks so much for your concern and suggestion. The names we used throughout the paper, were pseudonyms. We used pseudonyms because we promised to the IRB committee and participants not to reveal their names in anywhere of the paper. Please check line number 194-196 in the methodology section. 

Minor critique:

Comment Page 4 (line 89-91): sentence should be rephrased to increase clarity

Response: Thanks so much for your suggestion. We have made necessary edits for clarification. Please check line number 95 to 97 in the revised version. 

Comment Page 5 paragraph titled "COVID-19 situation and response in Bangladesh' seems oddly placed. Consider moving it prior to the previous paragraph (lines 96-104)

Response: Thank you so much for your suggestion. Given the long introduction section with an extensive literature on AMR problem globally and LMICS settings, we wanted to provide a better understanding to the readers on twin burdens of COVID-19 and AMR. Then we wanted to give a background on the country context of COVID-19 and Government’s response. This section will help readers to correlate our results on structural drivers, specifically, how health systems might improvise participants’ decision to take antibiotics. Therefore, we want to stick on the current form. 

Comment Page 6 line 134: Is that the total number of COVID cases in Bangladesh at the time? Please clarify. Also 'Residents both in cities' is a typo.

Response: Thanks so much for your comment. Yes, according to the official statistics, 141,801 were total number of clinically confirmed cases at the time of data collection in Bangladesh. We have now made it clear. Please check line number 144-146 in the revised version. 

We have now replaced that 'Residents both in cities' with ‘potential participants’. Please check line number 146 in the revised version.  

Reviewer #2: This is a very interesting and well written qualitative study of antibiotic use among people with COVID-19 positive test or symptoms in Bangladesh. This was a fascinating read and I very much appreciate the authors’ work. I have a few comments mostly meant for clarification.

Response: Thanks so much for your overall enthusiasm to read the paper critically and providing feedbacks, we believe, that will help to improve its clarity. In the following, we have clarified and made necessary changes based on your comments, suggestions and questions. 

Comment 1. Introduction – It would be helpful to provide a citation for the statements in the first two sentences (lines 45-48).

Response: Thanks for your suggestion. We have provided citation for these statements in the revised draft. Please check line 48-51 (citation 1-3) in the revised version. 

Comment 2. Introduction, Line 55 – typo in “…related mortality” (rather than morality)

Response: Made corrections accordingly. Please check line 59 in the revised version. 

Comment 3. Study setting, Line 133 – might want to clarify that the 70.6% and 14.1% figures are % shares of total number of cases, not a population prevalence of COVID (e.g., 70% of the population of Dhaka does not have COVID at any given time). 

Response: Thanks so much for raising this confusion. We have now revised it. Please check line number 143 to 146 in the revised version. 

Comment 4. Study participants – was there a timeframe during which participants had to have tested positive for COVID and/or were symptomatic? 

Response: No, we did not put any timeline for that. Since the study was conducted between May to June and the pandemic officially started from March in Bangladesh, we can assume that, the participants included in the study had COVID-19 and suggestive symptoms between this timeframe.

Comment Rather than listing gender/age as an inclusion criterion, I would note separately that purposeful sampling was used (or something, whatever is accurate) to ensure an even distribution of gender and age.

Response: Thanks so much for your suggestion. We have made necessary changes in the methodology section. Please see line number 153 in the revised version.

Comment 5. The inclusion criteria state that taking antibiotics in the last 12 months was an inclusion criterion, but the table 1 characteristics state that 19 participants hadn’t taken antibiotics in the last year. Was past-year antibiotic use a criterion for inclusion in this study?

Response: Thanks so much for pointing out this. It was a typo. We have now made changes. Please see line number 150 in the revised version. 

Comment 6. I think Table 1 and associated results text should be in the results of the paper, not the methods. Additionally, it would be helpful to revise Table 1 into a more standard format with one row per characteristic (rather than two columns with characteristics), include both Ns and % of the population, etc. 

Response: Thanks for your suggestion. We have now moved table 1 down to the result section after making suggested changes. Please check in the result section. 

Comment In general, I would use standard section headings (Introduction, Methods, Results, Discussion) and sub-headings within each of those broader categories (e.g., for Study Participants).

Response: Necessary changes have made accordingly throughout. 

Comment 7. Were there participants in the sample who had not taken antibiotics for COVID? Were there any themes or results emerging from these interviews?

Response: Thanks so much for your question. As we wanted to explore social drivers of antibiotic consumption among individuals who had been tested positive with COVID-19 and had suggestive symptoms of COVID-19, we did not include those who had COVID-19 and suggestive symptoms but did not consume antibiotics. By inclusion criteria, we excluded these individuals. 

 

Reviewer #3: 

Comment - The title of the manuscript should be edited to better reflect the outcomes of this study. What do you mean by "Symptoms" in the title? It should be COVID-19-like symptoms...

Response: Thank you very much for your suggestion. The title of the manuscript has been modified and changed accordingly. Please check the revised version. 

Comment - The English writing of the manuscript needs proofreading.

Response: Thanks so much for your suggestion. The draft has been reviewed by two native English speakers who are authors (Dr Narelle Warren and Dr Mark Davis – affiliated with Monash University, Australia) in the paper.

Comment - How the authors diagnosed COVID-19 infection? This should be clearly explained. 

Response: Thanks so much for your suggestion. We have explained it in the methodology section. Please check line number 156-160 in the revised version. 

Comment - What types of antibiotics were consumed by patients? and for how long?

Response: Thanks so much for your questions. Prior to enrolling the participants, the participants were asked to mention the name of antibiotic they consumed. It was asked as part of inclusion criteria. As the main objective of the current paper is to offer social drivers of antibiotic consumption, we decided not focus on adherence behaviours in this paper. Moreover, given the diversified practices on adherence of practices, we may develop another paper on adherence. 

Comment - Were there any secondary bacterial infections? if yes, which infections?

Response: Thanks so much for your question. Unfortunately, we did not want to understand whether they had secondary infection or not. However, the secondary infection was brought up by the participants. What we mentioned in the draft was derived from the participants’ narratives, mostly from their perception. Since by criteria, all of the participants took home care, there was no clinical evidence on the type of secondary infection. As our main purpose was to explore the social drivers, we reported it as perceived secondary infection. 

Comment - The format of Tables needs edition.

Response: We have made necessary edits on the tables. Please check the revised version. 

Comment - The structure of manuscript is somehow strange. I do not know it is in the format of journal. For example, findings instead of Results!!!

Response: The structure of the draft has been changed throughout. 

Comment - The limitations of the study should be mentioned in the discussion.

Response: The limitation section was as part of the discussion section in the primary draft. However, we have specified it. Please check line number 537-547.

Comment- The following article fully explained the AMR situation during COVID-19 pandemic. Cite it in the Introduction or Discussion part.

doi: 10.3389/fmicb.2020.590683

"Antimicrobial Resistance as a Hidden Menace Lurking Behind the COVID-19 Outbreak: The Global Impacts of Too Much Hygiene on AMR"

Front Microbiol. 2020; 11: 590683.

Response: Thanks for your suggestion. This is worth citing. We have cited it in the Introduction section. Please check citation number 2. 

6. PLOS authors have the option to publish the peer review history of their article (what does this mean?). If published, this will include your full peer review and any attached files.

Do you want your identity to be public for this peer review? For information about this choice, including consent withdrawal, please see our Privacy Policy.

Reviewer #1: No

Reviewer #2: No

Reviewer #3: No

---

## [Decision Letter · Decision Letter 1]

1 Dec 2021

Understanding the Social Drivers of Antibiotic Use During COVID-19 in Bangladesh: Implications for Reduction of Antimicrobial Resistance.

PONE-D-21-18370R1

Dear Dr. Kalam,

We’re pleased to inform you that your manuscript has been judged scientifically suitable for publication and will be formally accepted for publication once it meets all outstanding technical requirements.

Kind regards,

Monica Cartelle Gestal, PhD

Academic Editor

PLOS ONE

Additional Editor Comments (optional):

Reviewers' comments:

Reviewer's Responses to Questions

**Comments to the Author**

1. If the authors have adequately addressed your comments raised in a previous round of review and you feel that this manuscript is now acceptable for publication, you may indicate that here to bypass the “Comments to the Author” section, enter your conflict of interest statement in the “Confidential to Editor” section, and submit your "Accept" recommendation.

Reviewer #2: All comments have been addressed

Reviewer #3: All comments have been addressed

2. Is the manuscript technically sound, and do the data support the conclusions?

Reviewer #2: Yes

Reviewer #3: Yes

3. Has the statistical analysis been performed appropriately and rigorously? 

Reviewer #2: N/A

Reviewer #3: I Don't Know

4. Have the authors made all data underlying the findings in their manuscript fully available?

Reviewer #2: Yes

Reviewer #3: Yes

5. Is the manuscript presented in an intelligible fashion and written in standard English?

Reviewer #2: Yes

Reviewer #3: Yes

6. Review Comments to the Author

Reviewer #2: (No Response)

Reviewer #3: The authors made the required corrections. It can be suitable for publication.

Good luck with your paper

7. PLOS authors have the option to publish the peer review history of their article (what does this mean?). If published, this will include your full peer review and any attached files.

Reviewer #2: No

Reviewer #3: No

---

## [Editor Report · Acceptance letter]

3 Dec 2021

PONE-D-21-18370R1 

Understanding the Social Drivers of Antibiotic Use During COVID-19 in Bangladesh: Implications for Reduction of Antimicrobial Resistance. 

Dear Dr. Kalam:

I'm pleased to inform you that your manuscript has been deemed suitable for publication in PLOS ONE. Congratulations! Your manuscript is now with our production department. 

Kind regards, 

on behalf of

Dr. Monica Cartelle Gestal 

Academic Editor

PLOS ONE